# Molecular Epidemiology of Mitochondrial Cardiomyopathy: A Search Among Mitochondrial and Nuclear Genes

**DOI:** 10.3390/ijms22115742

**Published:** 2021-05-27

**Authors:** Cristina Mazzaccara, Bruno Mirra, Ferdinando Barretta, Martina Caiazza, Barbara Lombardo, Olga Scudiero, Nadia Tinto, Giuseppe Limongelli, Giulia Frisso

**Affiliations:** 1Department of Molecular Medicine and Medical Biotechnology, University of Naples Federico II, 80131 Naples, Italy; mirrab@ceinge.unina.it (B.M.); barretta@ceinge.unina.it (F.B.); barbara.lombardo@unina.it (B.L.); olga.scudiero@unina.it (O.S.); nadia.tinto@unina.it (N.T.); gfrisso@unina.it (G.F.); 2CEINGE Advanced Biotechnologies, 80145 Naples, Italy; 3Monaldi Hospital, AO Colli, 80131 Naples, Italy; martina.caiazza@yahoo.it (M.C.); limongelligiuseppe@libero.it (G.L.); 4Department of Translational Medical Sciences, University of Campania “Luigi Vanvitelli”, 80134 Naples, Italy

**Keywords:** mitochondrial DNA, mitochondrial cardiomyopathy, mitochondrial disease, next generation sequencing, genetic testing, diagnosis, mutation

## Abstract

Mitochondrial Cardiomyopathy (MCM) is a common manifestation of multi-organ Mitochondrial Diseases (MDs), occasionally present in non-syndromic cases. Diagnosis of MCM is complex because of wide clinical and genetic heterogeneity and requires medical, laboratory, and neuroimaging investigations. Currently, the molecular screening for MCM is fundamental part of MDs management and allows achieving the definitive diagnosis. In this article, we review the current genetic knowledge associated with MDs, focusing on diagnosis of MCM and MDs showing cardiac involvement. We searched for publications on mitochondrial and nuclear genes involved in MCM, mainly focusing on genetic screening based on targeted gene panels for the molecular diagnosis of the MCM, by using Next Generation Sequencing. Here we report twelve case reports, four case-control studies, eleven retrospective studies, and two prospective studies, for a total of twenty-nine papers concerning the evaluation of cardiac manifestations in mitochondrial diseases. From the analysis of published causal mutations, we identified 130 genes to be associated with mitochondrial heart diseases. A large proportion of these genes (34.3%) encode for key proteins involved in the oxidative phosphorylation system (OXPHOS), either as directly OXPHOS subunits (22.8%), and as OXPHOS assembly factors (11.5%). Mutations in several mitochondrial tRNA genes have been also reported in multi-organ or isolated MCM (15.3%). This review highlights the main disease-genes, identified by extensive genetic analysis, which could be included as target genes in next generation panels for the molecular diagnosis of patients with clinical suspect of mitochondrial cardiomyopathies.

## 1. Introduction

Cardiomyopathies (CM) represent a wide group of human disorders, extending from isolated cardiac diseases to cardiac involvement in the context of complex diseases, such as neuromuscular disorders, mitochondrial pathology, and metabolic/infiltrative/storage disease [1,2]. An extensive clinical examination, comprising a systematic screening of several organs, should be, initially, conducted to define whether the cardiomyopathy is isolated or it is part of a multisystem disorder. Furthermore, the family history, specific symptoms, and the presence of typical clinical and instrumental findings may help the physicians in the differential diagnosis between mitochondrial cardiomyopathy (MCM) and sarcomere disease. Indeed, mitochondrial diseases (MDs) may manifest with structural and clinical features of hypertrophic (HCM), dilated (DCM), or restrictive (RCM) cardiomyopathy, in absence of concomitant coronary artery disease, valvular disease, or hypertension [3,4,5,6,7]. MDs are a group of rare disorders (prevalence 5–12/100,000), due to both mitochondrial DNA (mtDNA) and nuclear genome (nDNA) alterations, characterized by defect of oxidative phosphorylation (OXPHOS), impairing the ATP production [8,9,10,11,12] (Appendix A: mitochondrial disease).

MDs show an extremely clinical, biochemical, and genetical heterogeneous phenotype, depending on the involved tissue, the specific nuclear or mitochondrial DNA mutation, and mtDNA heteroplasmic level. Several degenerative, neuromuscular, ophthalmological and gastroenterological disorders, deafness, and endocrine manifestations [13,14,15,16], such as diabetes mellitus, are frequently observed in the affected individuals [11,13,17,18,19,20]. In MDs the heart, being a high-energy demand tissue, is one of the main affected organs, [21]. In fact, cardiac involvement in patients with MDs has been estimated to be approximately 20–25%, and up to 40% in children [19,22] and 30% in adults [23].

In mitochondrial cardiomyopathies, the biogenesis of mitochondria represents an adaptive response to the mitochondrial defects. In these pathologies, the energy deficiency, mechanical interference due to sarcomere misalignment and uneven contraction, increased oxidative stress, and uncoupled respiration are possible harmful factors to myocyte function [24].

In particular, HCM due to inter-myofibrillary proliferation of mitochondria and resulting in mechanical damage and deficit of sarcomere function, is the most common cardiac manifestation of MDs, occurring in approximately 40–50% of MCM cases [7,16,19]. Dilated or restrictive cardiomyopathy and cardiac conduction defects are less commonly identified in MDs [25].

Although some MDs may affect a single organ, most of them interest multiple organs, displaying a clinical syndrome (Appendix A: mitochondrial disease).

In many individuals, the diagnosis is complex and requires a multidisciplinary approach including clinical, biochemical, histochemical, immunocytochemistry, and neuropathological investigations. Finally, the molecular analysis, by mtDNA and nDNA sequencing, may reveal the genetic basis of the pathology. To date, besides the 13 OXPHOS system proteins, 22 tRNA, and 2 rRNA, overall encoded by the mitochondrial genome, about 1500 nuclear-genome encoded proteins, encompass the mammalian mitochondrial proteome. These proteins constitute the structural subunits of the OXPHOS system, are responsible of the mtDNA maintenance, replication, and function, as well as are involved in mitochondrial assembly and stability of the five OXPHOS complexes. [1,16,26,27]. Because of this great genetic heterogeneity, the MDs genetic testing, through the candidate gene sequencing approach, may not identify the molecular alteration, leaving many patients without diagnosis. In recent years, high-throughput sequencing strategies, such as next-generation sequencing (NGS), are proving to be fundamental for the diagnosis of MD patients and for mechanistic MDs studies [28,29,30].

In this article, we reviewed the current genetic knowledge on MDs, focusing on diagnosis and management of MCM and MDs showing cardiac involvement. We searched for publications on mitochondrial and nuclear genes involved in mitochondrial cardiomyopathies, and identified a number of targeted genes, which should be included in genetic-panel for the molecular diagnosis of the mitochondrial cardiomyopathies, by using Next Generation Sequencing.

## 2. Mutations in Nuclear and Mitochondrial Genes

Several variations in mitochondrial and OXPHOS-related nuclear protein coding genes cause a reduction in enzyme function in one or more of the RC complexes, while mt-tRNA mutations may impair mitochondrial translation, by decreasing the availability of functional mt-tRNAs [31]. These dysfunctions, overall, cause defects of the oxidative phosphorylation system and subsequently impair the ATP production, leading to different phenotypes, depending on the tissue involved, type of pathogenic mutations and heteroplasmy level [10,11]. Estimates suggest that about 15–20% of RC defects are due to mtDNA mutations, while the remaining are caused by defects in nuclear genes [19,32].

Mitochondrial mutations may be inherited (~75% cases) or may occur de novo (~25% cases) [33]. MtDNA point mutations represent the most frequent causes of mtDNA alterations [8,11]. MtDNA point mutations (including small indel mutations) are pretty frequent as they are found in 1:200 newborns; however many individuals carry low heteroplasmic level of the mutant gene, thus they remain asymptomatic [34]. On the contrary, large-scale mtDNA deletions are generally rare (1.5/100,000) and typically arise as de novo mutations during embryonic development. Conversely to nDNA rearrangements, large-scale mtDNA deletions have a low recurrence risk (<10% risk of transmission), although the identification of the same deletion in mother and son suggests a matrilineal transmission [35,36,37]. In a more recent classification, the primary mitochondrial diseases (PMD) are caused by germline mutations, in mtDNA and/or nDNA genes encoding RC proteins or in nDNA genes involved in the complex protein-organization, needed for mitochondrial RC [38]. Conversely, secondary mitochondrial dysfunction (SMD) refers to abnormal mitochondrial function caused by mutations in non-OXPHOS genes affecting the mitochondrial dynamics and its capacity to produce ATP. SMD can be inherited or acquired (due to non-genetic causes such as environmental factors) which is an important distinction from PMD, which can only be inherited.

Furthermore, in addition to ATP deficiency, mitochondrial dysfunction, contributing to pathogenesis of MDs, may also affect calcium handling, ROS production, apoptosis dysregulation and nitric oxide (NO) deficiency [39].

Although the production of ROS and accumulation of their active metabolites is reported as the major known cause of mtDNA mutations [40], recent studies have reported mitochondria fission and fusion (mitochondrial dynamics) imbalance in cardiomyopathies [41]. In particular, the processes of mitochondrial fusion, fission, biogenesis and mitophagy, involved in mitochondrial morphology, quality, and abundance, have recently been implicated in cardiovascular disease. Studies suggest that changes in mitochondrial morphology may be an adaptive mechanism during cardiac pathological remodeling [42].

Genetic deletions of mitochondrial fusion-promoting genes, Mitofusin 1 and 2 (*MFN1*, *MFN2*), have been shown to induce mitochondrial fragmentation, whereas deletions of mitochondrial fission-promoting gene (Dynamin-1-like *DRP1*) cause mitochondrial enlargement. Likewise, because the inner mitochondrial membrane fusion protein OPA1 (optic atrophy factor 1 *OPA1* gene) is essential for maintaining the shape of the mitochondria and normal cristae structure, cardiomyopathies, caused by *OPA 1* gene deletions may determine defective mitochondrial inner membrane architecture [43]. Chen et al. [44] described small and fragmented mitochondria in both human and rat models of heart failure, which were associated with decreased OPA1 levels.

Furthermore, nuclear gene mutations, regulating mtDNA replication and maintenance, such as mitochondrial transcription factor A (*TFAM*), mtDNA polymerase γ (*POLG*), and *PEO1* (Twinkle) genes have been associated with cardiac diseases [45,46].

## 3. Mitochondrial Cardiomyopathy (MCM)

The most frequent cardiac disorders refereed to mitochondrial dysfunction are cardiomyopathies [3,16,25]. Although the true prevalence of mtDNA-related CM is unknown, based on the prevalence of mtDNA disease and frequency of cardiac manifestations, about 1/10,000–15,000 of the general population is affected [7]. Furthermore, several studies report different prevalence between pediatric and adult population, with cardiomyopathy estimated to occur in 20–40% of children with MDs [16,22,47,48].

Mitochondrial cardiomyopathy is a myocardial disorder, without concomitant coronary artery disease, valvular disease, congenital heart disease, and hypertension, characterized by abnormal myocardial structure and/or function, subsequent to genetic alterations and resulting in the impairment of the mitochondrial respiratory chain [4]. Hypertrophic cardiomyopathy, dilated cardiomyopathy, and left ventricular non compaction (LVNC) are the main manifestations of MCM. Histiocytoid cardiomyopathy (HICMP) (Purkinje fiber dysplasia) and restrictive cardiomyopathy, as well as Takotsubo syndrome (TTS) have been also observed [16,19].

Moreover, MCM, in addition to the myocardium, may affect other cardiac tissue, such as the cardiac conduction system, cardiac valves, aortic root, coronary arteries or the pericardium, and connective tissue, thus leading impulse generation/conduction insufficiency, valvular diseases, systolic dysfunction, or heart failure and pulmonary arterial hypertension (PAH). Conduction system disorders arise in 5–10% of patients with MD, especially in patients with associated neuromuscular disease [49].

The abnormal heart-muscle structure and/or function is due to genetic defects, impairing the functionality of the mitochondrial RC complexes, their assembly and stability factors, tRNAs, rRNAs, mtDNA maintenance, and coenzyme Q10 (CoQ10) synthesis [19,50].

The cardiac involvement in MDs may be already evident at onset, but in some cases, it may develop over time. More rarely, in patients without or with mild multisystem involvement, the heart may be the only or the main clinically affected organ.

The imbalance of mitochondrial function in MCM is supported by a large amount of scientific publications, investigating the cardiological aspect both as one of the manifestations of a syndromic mitochondrial disorder and, in a smaller number of cases, as an isolated cardiac disease. In particular, Whabi et al. showed that myocardial involvement in mtDNA diseases is progressive and, in association with other cardiovascular manifestations and risk factors (i.e., intraventricular conduction block, diabetes, premature ventricular contractions, and left ventricular hypertrophy-LVH), is an independent predictor of morbidity and early mortality [23].

### 3.1. Hypertrophic Cardiomyopathy

Typical cardiac manifestation of MDs is HCM, occurring in about 50% of patients with MCM [22,51,52]. Generally, these subjects do not show left ventricular outflow tract obstruction and evolution toward left ventricular systolic dysfunction is more common than in sarcomere HCM [7,25]. Uncommonly, HCM appears as the only MDs manifestation, while frequently it’s part of a syndromic disorder, like mitochondrial encephalomyopathy with lactic acidosis and stroke-like episodes (MELAS) (OMIM 540000), myoclonic epilepsy with ragged-red fibers (MERRF) (OMIM 545000), Chronic progressive external ophthalmoplegia (CPEO) (OMIM 157640), Leber hereditary optic neuropathy (LHON) (OMIM 535000), or neurogenic weakness with ataxia and retinitis pigmentosa (NARP) (OMIM 551500), diseases [53].

Mutations in nuclear and mitochondrial genes encoding mitochondrial RC complex subunits and mitochondrial RC complex assembly factors have been associated with HCM [54,55,56,57,58]. Furthermore, mutations in nuclear TSFM, encoding a mitochondrial translation elongation factor, have been identified in HCM or DCM with multi-organ disease [59].

Point mutations in mtDNA, particularly in mt-tRNA genes, seem to be more frequently associated with hypertrophic phenotype.

Left ventricular hypertrophy has been identified in 38–56% of patients harboring the m.3243A>G mutation, in the tRNA for Leucine gene (MTTL), which is the most common mutation involved in several MDs, also reported in DCM patient [25].

Furthermore, heart failure with impaired systolic function and ventricular dilatation, leading sudden death has been reported in patients with LVH bearing the mutations m.3243A>G in MTTL gene or m.8344A>G in MTTK gene [4,60]. Sudden death may occur in patients bearing the m.3243A>G mutation also in absence of myocardial involvement [61].

Table 1 shows the main nuclear and mitochondrial gene mutations associated with different cardiological phenotype [62,63,64,65].

### 3.2. More Rare Cardiomyopathies

Dilated cardiomyopathy occurs more rarely in MD patients, in which DCM may represent the initial manifestation of cardiac involvement [66]. Furthermore, some studies found DCM more frequently associated with mitochondrial diabetes (MIDD, Maternally Inherited Diabetes and Deafness) (OMIM 520000) [67]. More commonly, the cardiac chamber dilation and systolic dysfunction represent the progression of pre-existing hypertrophy, reported in about 10% of HCM patients [68]. Mutations in tRNA or in genes encoding subunits of the mitochondrial RC complexes as well as single or large-scale mtDNA deletion and/or depletion have been observed (Table 1) [7,60,69]. Mitochondrial cardiomyopathy may also appear as RCM, LVNC, or HICMP [70,71].

Mutations in mtDNA are common in LVNC, whereas sarcomere or ion channel genes variants account for few LVNC cases [72]. LVNC is caused by defects during cardiac development with abnormal myofibrils compaction and may evolve towards progressive ventricular dilatation and systolic dysfunction. This disease is commonly reported as a cardiac manifestation in multisystem diseases, especially in pediatric populations [4,73,74]. Histiocytoid cardiomyopathy is another rare cardiomyopathy, showing severe cardiac arrhythmias or dilated cardiomyopathy and arising in infancy or childhood [75]. Autosomal recessive, X-linked, and maternal transmission have been described; nevertheless, several reports indicate that HICMP is a cardiac manifestation of MDs and should be considered as MCM [76,77,78]. Histiocytoid cardiomyopathy is characterized by infiltration of the myocardium by pathognomonic histiocyte-like cells with foamy or granular cytoplasm full of morphologically abnormal mitochondria, showing significantly decreased succinate-cytochrome C reductase (SCR) or NADH-cytochrome C reductase activity [78]. Known molecular defects associated with HICMP are in Table 1.

### 3.3. Mitochondrial DNA in Conduction System and Coronary Heart Disease

MDs are responsible for defects of the contractile cardiac system, leading, indirectly, to dysfunction in cardiac electrical function. Arrhythmogenic cardiac diseases, including atrioventricular (AV) blocks, ventricular and supraventricular arrhythmias, associated with pre-excitation syndrome, represent the most common abnormalities of the cardiac electric system in MDs. Reduced ATP synthesis and increased ROS production, caused by the mitochondrial dysfunction, can directly damage the cardiomyocyte excitability.

In patients with mitochondrial dysfunction, the reduced ATP synthesis leads to decreased ATP/ADP which results in constitutive opening of the ATP-sensitive potassium (KATP) channels on the sarcoplasmic membrane; this reduces electrical transmission, by creating current sinks in the myocardium and decreases refractory periods, both increasing electrical instability and consequently, the arrhythmic risk [79].

Conduction system diseases, occurring in 5–10% of MDs patients are reported in association with MTTL and MTTK gene mutations, (Table 1) [49,60,80].

Intraventricular conduction anomalies may be responsible for sudden cardiac death (SCD) in these subjects [53,81].

In MDs patients, progression to high-grade AV block is often unexpected; hence, a fast identification of any conduction system disease is required for a timely intervention. In these patients, pacemaker is usually proposed on early PR-interval prolongation or bundle branch block [82].

Furthermore, coronary heart disease (CHD) may be a manifestation of MDs. CHD characterized by atherosclerosis of epicardial coronary arteries leading to vessel stenosis. Severe ischemia may lead to myocardial infarction and angina pectoris [83]. Genetic alterations and environmental factors increasing ROS production as well as insufficient antioxidant mechanisms, concur to the pathogenesis of atherosclerosis [84]. Numerous studies show that mtDNA mutations, altered mitochondrial DNA copy number (mtDNA-CN) and specific mtDNA haplogroups are involved in the CHD and myocardial infarction [85,86,87,88,89]. In particular, Kofler et al., found that mitochondrial haplogroup T was associated with CHD in Middle European Caucasian populations [86]. Mitochondrial mutations involved in atherosclerosis development and its clinical manifestations are showed in Table 1. These mutations cause defects in some mitochondrial respiratory chain proteins and tRNAs, leading their reduced concentration with deficit of mitochondrial function, and contributing to the development of atherosclerosis [90,91].

### 3.4. Other Mitochondrial Diseases with Cardiomyopathy

Other MDs with cardiological manifestations include Barth syndrome (OMIM 302060), Sengers syndrome (OMIM 212350), TMEM70-related mitochondrial complex V deficiency (OMIM 614052), and Friedreich ataxia (OMIM 229300) [92].

Barth syndrome (OMIM 302060) is an X-linked disorder, whose most common cardiac manifestations are LVNC and DCM, while HCM is uncommon. Barth syndrome is caused by mutations in the gene coding for tafazzin (TAZ), which is an inner mitochondrial membrane phospholipid transacylase with key role in remodeling of cardiolipin [93].

Sengers syndrome (acylglycerol kinase deficiency) is characterized by 3-methylglutaconic aciduria. Patients show HCM, myopathy, congenital cataracts, lactic acidosis, and exercise intolerance. Heart failure causes death in half of patients within the first year of life. The syndrome is caused by mutations in *AGK* gene, coding an acylglycerol kinase involved in the assembly of the mitochondrial adenine nucleotide transporter ANT1 [94]. The transmembrane protein 70, encoded by *TMEM70* gene, is a protein located in the mitochondrial inner membrane involved in assembling and stabilizing of the mitochondrial respiratory chain Complex I and V proteins.

Alterations in TMEM70 cause Complex V deficiency but are reported also in the less severe Complex I deficiency and in combined OXPHOS deficiency [95].

Mutations in *TMEM70* gene (Table 1) are the most common cause of ATP synthase deficiency, resulting in a multi-organ mitochondrial disease, known as neonatal mitochondrial encephalo-cardiomyopathy, which is characterized by a large variety of symptoms, including 3-methylglutaconic aciduria, lactic acidosis, mitochondrial myopathy, and HCM [96]. The main known mutations in this gene are c.317-2 A>G, located in the splice site of *TMEM70* intron 2, which leads to aberrant splicing and loss of TMEM70 transcript, with most patients not surviving the neonatal period and the c.366A>T, (p.Tyr112Ter) resulting in Nuclear Type 2 Mitochondrial Complex V deficiency, showing HCM. The study of Cameron J.M. et colleagues on mitochondrial morphology in patients with mutations in this gene, revealed abnormal mitochondria, with whorled cristae and disrupted nucleoid clusters of mtDNA [97].

Finally, HCM is the cardiac manifestation of Friedreich ataxia, an autosomal recessive neurodegenerative disease caused by mutations in frataxin (*FXN*) gene (Table 1) encoding a mitochondrial iron-binding protein involved in the synthesis of the iron-sulphur (Fe–S) clusters (ISCs). ISCs are inorganic redox-active protein cofactors required for the activity of the OXPHOS complexes [98,99]. The absence of FXN protein debars the ISC-assembly complex and increases the iron deposition and its oxidation.

Cardiac dysfunction, mainly congestive heart failure or arrhythmia but also stroke, ischaemic heart disease, and pneumonia, are the most common causes of death in these patients [100].

**Table 1 ijms-22-05742-t001:** List of known pathogenic mutations reported in association with cardiological phenotype, by MITOMAP database, literature data and using ACMG classification.

Cardiological Phenotype	Disorder	Mutations	Amino Acid Change	Ref
**HCM**	MELAS, MERRF, CPEO, LHON, NARP, MIDD, Sengers syndrome, Friedreich ataxia	*ACAD9*: c.797G>A	p.Arg266Gln	[101]
		*AGK:* c.306T>G	p.Tyr102Ter	[102]
		*COX6B1*: c.58C>T	p.Arg20Cys	[56]
		*FXN*: GAA repeat expansion	-	[98]
		*MRPL3*: c.950C>G	p.Pro317Arg	[63]
		*MRPL44*: c.467T>G	p.Leu156Arg	[103]
		*MTCOX2*: m.7896G>A	p.Trp104Ter	[58]
		*MTCYB*: m.14849T>C	p.Ser35Pro	[104]
		*MTND1*: m.3481G>A	p.Glu59Lys	[105]
		*MTND5*: m.12338T>C	p.Met1Thr	[106]
		*MTRNR2*: m.2336T>C	-	[107]
		*MTTK*: m.8344A>G	-	[60]
		*MTTI*: m.4300A>G	-	[65]
		*MTTL*: m.3243A>G	-	[4]
		*NDUFS2*: c.686C>A	p.Pro229Gln	[108]
		*NDUFV2*: c.669_670insG	p.Ser224fs	[109]
		*NDUFA2*: c.208+5G>A	-	[110]
		*NDUFAF1*: c.631C>T	p.Arg211Cys	[111]
		*SCOX2:* c.418G>A	p.Asp140Asn	[112]
		*SURF1:* 845_846delCT	p.Ser282Cysfs	[113]
		*SDHD*: c.275A>G	p.Asp92Gly	[114]
		*TMEM70*: c.317-2A>G	-	[115]
		*TMEM70*: c.366A>T	p.Tyr112Ter	[116]
		*TSFM:* c.997C>T	p.Arg312Trp	[117]
**DCM**	MELAS, MIDD, LHON, Barth syndrome	*MTTL*: m.3243A>G	-	MITOMAP
		*MTTI*: m.4300A>G	-	MITOMAP
		*MTTK*: m.8344A>G	-	MITOMAP
		*MTND4*: m.11778G>A	p.Arg340His	MITOMAP
		*TAZ:* c.527A>G *TSFM:* c.355G>C	p.His176Arg	[118]
			p.Val119Leu	[119]
**RCM**	Hearing loss and multi organ mitochondrial disorder, MELAS, MIDD	*MTRNR1*: m.1555A>G	-	MITOMAP
		*MTTL*: m.3243A>G	-	MITOMAP
**LVNC**	MIDD	*MTND1*: m.3398T>C	p.Met31Thr	MITOMAP
**HCM/LVNC**	Leigh syndrome	*MTND1:* m.3697G>A	p.Gly131Ser	MITOMAP
**HICMP**	MERF	*MTTK*: m.8344A>G	-	MITOMAP
		*MTCYB*: m.15498G>A	p.Gly251Asp	MITOMAP
**Conduction system** **disease**	KKS, CPEO	*MTTL*: m.3243A>G	-	MITOMAP
		*MTTK*: m.8344A>G	-	MITOMAP
**CHD**	carotid atherosclerosis risk, HCM, Leigh syndrome, MELAS	*MTCYB*: m.15059G>A	p.Gly105Ter	MITOMAP
		*MTCYB*: m.15243G>A *MTND5*: m.13513G>A	p.Gly166Glu	MITOMAP MITOMAP
			p.Asp393Asn	
		*MTTL1*: m.3256C>T	-	MITOMAP

CPEO: Chronic progressive external ophthalmoplegia; CHD: Coronary heart disease; DCM: dilated cardiomyopathy; HCM: Hypertrophic Cardiomyopathy; HICMP: Histiocytoid cardiomyopathy; KKS: Kearns–Sayre syndrome; LHON: Leber hereditary optic neuropathy; LVNC: left ventricular non compaction; MELAS: mitochondrial encephalomyopathy with lactic acidosis and stroke-like episodes; MERRF: myoclonic epilepsy with ragged-red fibers; MIDD: Maternally Inherited Diabetes and Deafness; NARP: neurogenic weakness with ataxia and retinitis pigmentosa; RCM: restrictive cardiomyopathy. All the variants are reported as pathogenetic by using the American College of Medical Genetics and Genomics (ACMG) interpretations [120] and/or reported in MITOMAP (A Human Mitochondrial Genome Database. Available online: https://www.mitomap.org/; accessed on 2 October 2020), Clin Var (Clinical Variants Database. Available online: https://www.ncbi.nlm.nih.gov/clinvar/; accessed on 2 October 2020), HGMD (The Human Gene Mutation Database Professional. Available online: http://www.hgmd.cf.ac.uk/; accessed on 2 October 2020) databases and literature data.

### 3.5. Multiorgan Clinical Expression of Mitochondrial Cardiomyopathy

The clinical expression of MCM is often accompanied by multisystem manifestations presenting neuromuscular, endocrine, and neuro-sensorial features [121,122]. Most patients with neuromuscular signs show creatine kinase enzyme in reference values or slightly elevated levels, while higher liver enzyme levels have been found in up to 10% of patients. Renal features may include nephritic syndrome, tubulopathy, tubulointerstitial nephritis, and nonspecific renal failure. Endocrinopathies include hypothyroidism, hypoparathyroidism, diabetes mellitus, adrenocorticotropic hormone deficiency, and hypogonadism. Gastrointestinal symptoms (diarrhea, constipation, abdominal pain, nausea, and chronic intestinal pseudo-obstruction) may also be present. The main ophthalmologic manifestation is retinitis pigmentosa. Sensorineural hearing loss occurs in 7% to 26% of patients, and its prevalence increases with age [4].

## 4. Diagnosis

Diagnosis of MCM is particularly complex because of wide clinical and genetic heterogeneity and requires a multidisciplinary approach, including extensive medical, laboratory and neuroimaging investigations. Physicians should suspect a possible MCM in patients with signs and symptoms of cardiomyopathy and multisystem involvement without a clear cause and refer the patients to a geneticist or other specialists with MDs expertise.

Genetic counselling is a fundamental part of the diagnostic workup and should be performed by specialized personnel. A detailed pedigree of the patient’s family should be obtained, and all first-degree relatives should be referred to a specific cardiogenetic clinic. An integrated diagnostic approach, consisting of biochemical screening, histopathological studies, cardiological investigations, including magnetic resonance imaging, functional assays, and molecular genetic testing, should be conducted to reach a correct diagnosis. The identification of a causative mutation allows the presymptomatic identification and the follow up of family members. The advent of NGS techniques has revolutionized the diagnosis of MDs in terms of expanding the number of involved genes as well as the discovery of new genes potentially associated with the diseases.

### 4.1. Medical Examination

An initial careful clinical examination of several organs (including the evaluation of extra-cardiac features, i.e., diabetes and deafness) with different criteria for adults or children should initially be performed to determine whether the CM is an isolated pathology or part of a multisystem disease [4,16,121,123,124]. Cardiologists should suspect a mitochondrial dysfunction in patients presenting, in addition to cardiomyopathy, also hearing loss, renal dysfunction, diabetes mellitus, and peripheral myopathy.

A maternal inheritance pattern or family history of mitochondrial disease may support the suspicion of MCM [125].

### 4.2. Laboratory Investigation

As above reported, the laboratory investigations of suspected MD are very complex, and evaluation of selected biomarkers is necessary to guide genetic investigation. Recently, the Mitochondrial Medicine Society has provided consensus statements on the diagnosis and management of MDs [126,127].

Functional assays on tissue (typically skeletal muscle) based on measurement of RC complexes activity and mitochondrial oxygen consumption have been fundamental in the diagnosis of MDs, especially prior to the recent developments in genomics and remain important procedures to evaluate the mitochondrial function.

Although MDs present with a wide spectrum of clinical manifestations, the skeletal muscle is frequently affected and represents an excellent post-mitotic tissue with distinctive histological and histochemical hallmarks of mitochondrial pathology (in primary mtDNA-related disease). The typical presence of ragged-red fibers (RRF), by using the modified Gomori trichrome stains, represents the histologic hallmark of MDs, resulting from a compensatory response to a RC biochemical defect, although these fibers are typically absent in children and in many adults are present only in late-stage disease [4,8]. Furthermore, the histochemical activities of the partially mtDNA-encoded cytochrome C oxidase complex IV (COX) and the fully nuclear-encoded succinate dehydrogenase complex II (SDH) are measured in the diagnosis of adult-onset MDs [127].

The dual COX/SDH stain is used to determine whether mitochondrial anomalies, observed on Gomori trichrome stains, (and Hematoxylin and Eosin stain) are due to mtDNA or nDNA alterations. Mutations in mtDNA impair mainly synthesis of mtDNA-encoded complex IV subunits, resulting in a decreased or absent complex IV activity and the affected fibers show either partial or complete lack of brown staining (the so-called “COX-negative fibers”).

In contrast, SDH staining depends on the enzymatic activity of the Complex II, which is entirely encoded by nDNA. This complex maintains a normal activity even when mtDNA mutations are present [8,128].

The more recent quadruple immunofluorescence technology, using fluorescently labelled antibodies against subunits of complexes I–IV, provides accurate data on the relative amount of complex I and complex IV and could contribute to the diagnosis of MDs [129].

Furthermore, electron microscopy of the affected muscle biopsy can sometimes highlight an increased number of mitochondria with variable size and shape. In particular enlarged and swollen mitochondria, with irregular cristae and paracrystalline inclusions, are showed [130]. This mitochondrial morphology reflects the perturbation of the rates of fission and fusion, essential processes for the healthy maintenance of mitochondria, leading to either fragmented, punctiform mitochondria, or excessively long or interconnected mitochondria [131].

### 4.3. Cardiological and Neuroimaging Investigations

As above reported, all the major cardiomyopathy phenotypes may be shown in MDs, but hypertrophic remodeling is the dominant pattern of cardiomyopathy in MDs.

The early stages of MCM are characterized by progressive diastolic dysfunction and heart failure with preserved ejection fraction [125]. The association of LV hypertrophy (with or without apical trabeculation) with systolic dysfunction has been reported as a typical evolution of MCM. T-wave abnormalities are an ECG sign of MD, even in absence of LV hypertrophy. In a cohort of 32 MDs subjects, Limongelli et al. found an abnormal ECG in up to 68% patients [53]. Furthermore, the presence of a short PR interval, various degrees of AV block or Wolff–Parkinson–White syndrome, which is common in patients with mtDNA mutations, also supports the diagnosis.

Regarding cardiac magnetic resonance findings, characteristic patterns of cardiac association might be shown in some MDs, as reported by Florian A. et colleagues [132]. For example, concentric LV hypertrophy, with intramural late gadolinium enhancement (LGE) in the inferolateral wall may be considered a distinctive feature in patients with KSS [25].

## 5. Molecular Analysis of Mitochondrial Cardiomyopathies

Traditionally, for different pathologies as well as MDs, including MCM, molecular diagnosis is achieved through “candidate gene” studies. This approach is based on recognizable clinical manifestations and family history, followed by molecular testing performed by Sanger sequencing of known genes [133]. Indeed some MD phenotypes are linked to molecular defects in one or few mitochondrial and nuclear genes [12].

To date, for patients suspected of having maternally inherited mtDNA disorders, the molecular diagnosis is still initially based on the mtDNA Sanger sequencing. In adult onset cases, mtDNA aetiology is recurrent and mtDNA sequencing for point mutations or the whole mitochondrial genome, when common point mutations are not detected, remains the first approach [134].

Moreover, Sanger sequencing does not detect either heteroplasmic mutations below 15–20% or allow measuring the degree of heteroplasmy. It also does not show large deletions, which are, conversely, detectable by other different technologies (i.e., array comparative genome hybridization) [135,136,137].

The genetic diagnosis of many MDs, including MCM, is further impaired by numerous nuclear genes whose defects may cause mitochondrial diseases [28].

Nowadays, for the above mentioned reasons, the diagnosis of MCM based on clinical/biochemical and instrumental examinations and sequencing of candidate genes one by one, is not a practical approach because it leaves many undiagnosed cases.

Advances in genomics and high-throughput sequencing technologies, such as NGS, have revolutionized the diagnostic setting of MDs, allowing identification of a large number of nuclear genes simultaneously, but also providing the identification and an accurate measure of heteroplasmy, in case of mtDNA mutations [29,30,138,139,140,141,142,143]. For making diagnosis of many complex diseases, characterized by a wide clinical spectrum and heterogeneous genetic involvement, next generation sequencing based approach is available and sometimes necessary [144,145].

NGS screening encompasses different configurations, ranging from a small number of genes (targeted gene panels: i.e., Complex I genes) to hundred or more genes (targeted exome sequencing: i.e., Mito Exome, that includes about 1500 genes related to mitochondrial structure and function), and even up to Whole Exome Sequencing (WES) [122,146,147]. The choice depends by clinical, laboratory, and instrumental data, based on the clinical indication, family history, biochemical, and histopathological findings and imaging studies. The costs, time, acceptable depth of coverage and the management of off-target findings have to be also evaluated.

### Variant Interpretation

The main difficulty in MDs molecular diagnosis, including MCM, is to distinguish rare or novel causative mutations from non-pathogenic polymorphisms in known/unknown genes [29].

Interestingly, rare variants may be also common in isolated genetic groups such as pathogenic mutations, which may show a higher carrier frequency within isolated populations due to the founder effect.

In the last decade, large-scale sequencing projects, integrating data from large population, have improved enormously the variants interpretation. In this context, exome data from different sequenced patients and healthy controls of the same population may contribute to assess the pathogenic role of the variant detected in the patient.

The databases of known nuclear/mitochondrial pathogenic mutations publicly accessible to define the MDs/MCM genetic diagnosis are listed in Appendix A: online database.

Moreover, the correct classification of novel variants needs functional evidence, conferred by studies often complicated to perform and requiring specific laboratory competences [148,149]. Conversely, in silico evaluations, using several bioinformatics tools, are used to predict the likelihood of pathogenicity of missense and splicing variants. For the interpretation of missense variants the most used bioinformatics tools are: PolyPhen (Polymorphism Phenotyping. Available online: http://coot.embl.de/PolyPhen; accessed on 2 October 2020), SIFT (Sorting Intolerant From Tolerant. Available online: https://sift.bii.a-star.edu.sg/; accessed on 2 October 2020), PMut (Molecular Modeling and Bioinformatics Group. Available online: http://mmb2.pcb.ub.es:8080/PMut/; accessed 2 October 2020). Furthermore, NetGene2 (Center for Biological Sequence Analysis. Available online: http://www.cbs.dtu.dk; accessed on 2 October 2020), ESE finder (Exonic splicing enhancer. Available online: http://rulai.cshl.edu; accessed on 2 October 2020) are investigate to predict splice site mutations.

Guidelines from the American College of Medical Genetics aim to standardize variants interpretation classifying them as “pathogenic”, “likely pathogenic”, “uncertain significance variant” (VUS), “likely benign”, and “benign” considering simultaneously the weight of different criteria, also including in silico analysis [120,150].

However, the significance and the possible consequences of VUS, often detected by NGS technologies, are difficult to establish. Indeed, VUS interpretation and their consequent clinical management is a major challenge following NGS-based molecular testing. In this framework, an important step in the molecular MDs diagnostic approach is to investigate the family segregation of the detected variant in comparison with the clinical phenotype within the family. The detection of the same variant in affected family members, as well as in unrelated individuals with similar phenotype, may provide a strong additional data to support its possible pathogenic role.

Furthermore, interpretation of mtDNA variants for MDs, including MCM, should consider heteroplasmy level, maternal inheritance, and typical clinical manifestations as neuropathy and hearing loss, which provide accurate information for genetic counselling.

## 6. Literature Review

We reviewed and collected the literature data from July 1990 to October 2020, (one papers published in January 2021) by searching publications including genetic data from patient’s cohort showing mitochondrial cardiomyopathy or mitochondrial disorder with cardiac involvement. Database searching and Inclusion/Exclusion Criteria are reported in Appendix A: online database. The objective was to investigate the genotype/phenotype correlation in several studies, including patients with syndromic and non-syndromic features of mitochondrial cardiomyopathy. For this aim we investigated (accessed on 2 October 2020) the most known databases, available online.

Furthermore, in recent years, especially since the introduction of NGS technology, several studies, using extensive genetic analysis, highlighted the main disease-genes associated with MCM and MDs showing cardiomyopathy; this has contributed to increase our knowledge on the molecular background of these diseases.

The second aim of this review was to identify disease-genes, by searching publications of the last ten years, in order to prove the utility of genetic tests in MCM diagnosis in patients with syndromic or non-syndromic MDs and to identify a targeted gene panel for the molecular diagnosis.

### Summary of Evidence

We found twelve case reports, four case-control studies, eleven retrospective studies, and two prospective studies for a total of twenty-nine papers (Table 2).

Due to the number of studies identified, it was not possible to describe each of them individually. However, in the Table 2, we show the type study (case-control, case report, retrospective study, prospective study) and briefly the aim of study for each paper. Although with different inclusion criteria, ranging from genotype data (i.e., presence of specific mutation) or phenotype manifestations (i.e., mitochondrial disease, cardiomyopathy, respiratory chain disease) all the selected studies report the number of patients with cardiac phenotype and phenotype/genotype correlation.

Before the advent of NGS technology, few of the genes and mutations were known to be associated with mitochondrial diseases. For example, (as also showed in Table 2) the MT-TL1 m.3243A>G was one of the main mutations known to cause MDs (i.e., MELAS and MIDD phenotype). The use of next generation sequencing technology has revolutionized the MDs diagnosis, leading to the discovery of about half of the overall three hundred currently known genes. Thus, by searching publications regarding the most recent advances in understanding the molecular genetic basis of mitochondrial cardiomyopathy during the last ten years [7,16,19,48,151,152,154,155,174,175,176] and investigating the above mentioned databases, we identified the main nuclear and mitochondrial genes reported in association with mitochondrial cardiomyopathy and MDs showing cardiac manifestations or conduction defects.

Table 3 shows the 130 genes (mtDNA and nDNA genes) collected according to their function showed by colored squares, as reported in Figure 1. This Table can thus represent a gene panel, from literature and database search, performed for targeted genes selection and could be used for the molecular screening of patients with MCM suspect.

## 7. Discussion

CMs are reported by the European Society of Cardiology (ESC) Working Group on Myocardial and Pericardial Diseases as structurally and functionally myocardial disorders, in the absence of coronary artery disease, hypertension, valvular heart disease, and congenital heart disease explaining the myocardial defect [177].

The contribution of clinical molecular biology in the evaluation and prevention of cardiological risk, in particular sudden cardiac death, and in the therapeutic approach [178], both in patients and in healthy subjects and athletes, is an important goal that involves many areas of the medical profession [179,180,181,182,183].

MCM, mainly HCM followed by DCM and LVNC or, more rarely, RCM and HICMP, are common cardiac symptoms of MDs and should be considered in patients with multi-organ involvement. Cardiac abnormalities, such as cardiomyopathy, arrhythmias, and conduction defects are, indeed, present in up to 60% of patients with MD and represent often a major determinant of morbidity and mortality [53]. The wide genetic heterogeneity associated with mitochondrial cardiomyopathies, supports the opinion that the “cardiological phenotype” should often be considered as a common feature of many mitochondrial disorders and more rarely may be the first, or even the only, clinical manifestation.

Moreover, because MCM may also occur in non-syndromic MDs or in patients with only mild multisystem involvement and the differential diagnosis with non-mitochondrial cardiomyopathy through cardiological and imaging investigations is often difficult a considerable number of MCM remain undiagnosed or delayed [125]. An early recognition of MCM is crucial to initiate therapeutic procedures avoiding heart failure and arrhythmias as well as mitochondrial crisis.

Currently, the molecular screening for MCM, is a fundamental part of the management of MDs that, through an integrated multidisciplinary diagnostic approach allows getting to the diagnosis.

Recently, the Mitochondrial Medicine Society has provided consensus statements on the diagnosis and management of MDs, including measurements of mitochondrial biomarkers in plasma, urine, and spinal fluid [126].

Moreover, in patients with suspected of MD, biochemical screening, as well as histopathological studies and detailed clinical/cardiological evaluation should be followed by genetic investigation to make a definitive diagnosis.

Molecular analysis of mtDNA may be, initially performed when a clear maternally inheritance is observed, while both the mtDNA and nDNA should be considered if the inheritance pattern is uncertain.

The first objective of this review was to identify the major genes frequently associated with mitochondrial cardiomyopathies, in order to propose, by using Next Generation Sequencing technology, a novel targeted gene panel for the molecular diagnosis of the diseases. For this purpose, we explored literature and public data concerning the cardiac manifestations in mitochondrial diseases/disorders, in order to investigate the genotype/phenotype correlation in patients with MCM. This data collection allowed us to extract twenty-nine papers showing patients with mitochondrial cardiomyopathy and mtDNA and/or nDNA mutations.

The most of these papers report mtDNA mutations in association with the cardiological phenotype, in agreement with the higher frequency, in adults, of mutations in the mitochondrial genome. Moreover, many of these studies are not carried out with high-throughput genetic technologies that are able to highlight several disease-genes associated with MCM and MDs showing cardiomyopathy.

In this context, the next-generation sequencing technology has allowed, for several diseases, a huge expansion of the diagnostic screening, increasing the spectrum of involved genes and suggesting new genotype/phenotype correlations [184]. This correlation is particularly important in the diagnosis of MDs, including MCM and for the identification of pre-symptomatic family members to take in follow up.

The actual NGS strategies allow screening patients with suspected MD through a wide range of approaches, including targeted gene panel sequencing, mitochondrial gene panel, clinical exome panel and Whole Exome Sequencing (WES). The choice of the technique depends on several factors, such as funding, bioinformatics, and laboratory expertise.

WES or clinical exome panel may be considered for group of genetic disorders, which may be examined following the same pipeline. Then, the use of virtually assembled gene panel (i.e., Mitochondrial Cardiomyopathy gene panel, metabolic gene panel, etc.) allows to select only data in genes of interest and filtering for further genes or exome analysis, in a subsequent bioinformatics re-evaluation (i.e., if, in the selected panels, no mutations are found and the clinical suspicion is further deepened and confirmed). This is very important because the MD phenotype overlapped with other metabolic and neuromuscular disorders.

Although coverage for WES is constantly increasing and may be generally comparable to that obtained by targeted gene panel analysis, challenges in obtaining sufficient coverage are due to the need, for economic reasons, to collect a greater number of samples in the same analytical session.

Furthermore, the most challenging aspect of this approach is given by the large amount of unknown identified variants or novel disease genes, which need further investigation both by in silico analysis and through functional studies [185]. In this regards, the role of molecular biology research is fundamental to expand the list of pathogenic variants and increase our knowledge about MCM.

Conversely, smaller or targeted gene panels, such as the one proposed in this study, achieve a high diagnostic yield, at lower operating costs, in particular when they are designed to target the high yield genetic content, or most relevant and clinically associated genes.

This is an ideal NGS approach for the diagnosis of heterogeneous disorders, like MDs, involving dual genome.

In the post-genomic era, the benefits of NGS in molecular diagnosis of rare diseases are well established. However, the use of NGS-based approach in diagnostics is increasing the yield of VUS. The significance and possible consequences of these variants are difficult to establish. The possibility to study a specific VUS, in the context of a large pedigree by analyzing the variant’s segregation, as compared with the clinical phenotype may be useful; however, it is often unfeasible. Additionally, when clinical features suggest an underlying disease, the presence of a VUS may deserve reinterpretation regarding its pathogenicity during follow-up according to accumulating literature knowledge. We believe that the interpretation of variants should be verified approximately every 6 months. Indeed, it is not uncommon that the interpretation of variants changes after further studies are published over time. Moreover, we are fully aware that in silico predictions are not enough to establish the variants’ pathogenicity and in vitro functional studies would be needed to have more conclusive information on each identified variant.

Mitochondrial cardiomyopathies are very rare disorders; in this setting, despite laboratory capacity to generate sequence data has greatly increased, with the advent of high-throughput next generation sequencing, the capacity to interpret genomic data and to identify disease genes can be difficult. The literature review highlighted about thirty high-confidence genes known to cause mitochondrial diseases in large cohorts of MCM patients. At the same time, we selected many other genes rarely reported associated with MCM (<1% of cases, each), which are reported as potential disease-genes based on known gene function, segregation with disease in families, computational analysis, etc., although more information is needed so that they can be classified as clinically actionable genes. Consequently, the probability of finding pathogenic variants is higher in high confidence genes than in low confidence ones, using the updated guidelines for the clinical interpretation of sequence variants developed by the American College of Medical Genetics and Genomics (ACMG).

Employment, in a cohort of patients showing clinical features of MCM, of the genes panel designed by carefully selecting genes previously associated with MCM, will yield rapidly actionable results (variants in high-confidence genes) and also identify potentially causal variants (in low-confidence genes) that will require further information to be clinically actionable.

Although a lot of work is still needed to easily and quickly diagnose a Mitochondrial Cardiomyopathy, a significant number of undiagnosed patients may be identified by NGS technology.

In this review, we identified 130 disease-genes associated with MCM, by searching molecular data from papers published in the last ten years, reporting new sequencing technologies. These genes could be included as target genes to build a novel gene panel for the molecular diagnosis of patients with clinical suspect of MCM.

## 8. Conclusions

Mitochondrial cardiomyopathy is a challenging diagnosis, requiring wide medical, neuroimaging, and laboratory examinations. The cardiologists have to keep the degree of suspicion high, looking also for signs from other tissues involved and the family history. Echocardiography and neuroimaging findings are variable, but the hypertrophic cardiomyopathy is the most common phenotype. Moreover, the increasing use of Next Generation Sequencing technologies allows collecting more genetic data, continually improving the diagnostic success. This review collects the major genes associated with mitochondrial cardiomyopathy in recent decades, underlining how the link between research and diagnosis is crucial to expand the list of pathogenic variants and concomitantly enhance our knowledge of mitochondrial cardiomyopathy.

## 9. Limitation

This review has some limitations concerning both the possibility that the filters utilized for the literature and database research, the keywords used, and their association may not have been enough to avoid losing papers. Furthermore, we did not differentiate pediatric and adult studies. Finally, the panel of genes selected from such a large literature will have to be validated to know the diagnostic yield in mitochondrial cardiomyopathies.

## Figures and Tables

**Figure 1 ijms-22-05742-f001:**
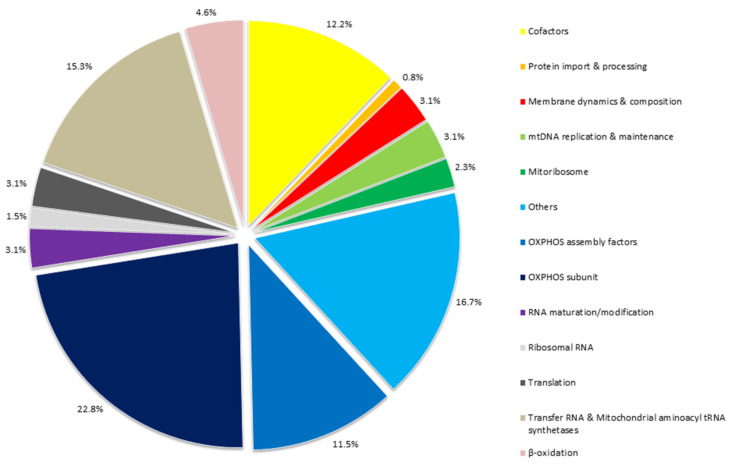
Rates of protein functions and RNA involved in mitochondrial cardiomyopathies. The figure groups all the 130 genes, reported in Table 3 associated with mitochondrial heart disease, according to their function. A large proportion of these genes (34.3%) encode for key proteins in oxidative phosphorylation system, either as directly OXPHOS subunits (22.8%) and as OXPHOS assembly factors (11.5%). Furthermore, mutations in several mitochondrial tRNA genes have been reported with multi-organ or isolated MCM (15.3%). Mainly HCM, but also DCM or histiocytoid cardiomyopathy, are the main cardiomyopathies associated with pathogenic variants in genes encoding mitochondrial tRNAs. Mutations in mt-tRNA may damage overall mitochondrial translation by alteration of functional mt-tRNAs.

**Table 2 ijms-22-05742-t002:** Summary of literature data concerning the evaluation of cardiac manifestations in mitochondrial diseases/disorders.

Authors	Type of Study	Aim of the Study	Inclusion Criteria	RESULTS			Ref.
				N. of Patients	N. of Patients with Cardiac Phenotype	N. of Patients with Positive Genotype ^§^ and MCM	
SP. Correia et al. 2021	*CR*	To investigate a patient with congenital lactic acidosis and HCM	Patients with HCM, lactic acidosis	1	1	1	[151]
H. Chung et al. 2020	*RS*	To investigate phenotype-based clinical and genetic characteristics of HCM patients using comprehensive genetic tests and rare variant association analysis	Patients with HCM	212	212	41	[152]
EMC. Chau et al. 2020	*CR*	To investigate MCM	MD phenotype	1	1	1	[153]
A. Imai-Okazaki et al. 2019	RS	To investigate CM in children with MD	MD genotype	137	29	29	[48]
R. Kamps et al. 2018	*CR*	To identify gene defects in pediatric CM and early-onset brain disease with OXPHOS deficiencies	Patients with CM	3	3	3	[154]
RG. Feichtinger et al. 2017	*CR*	To investigate by NGS four unrelated patients	Patients with combined RCD	4	4	4	[155]
K. Wahbi et al. 2015	*RS*	To assess the long-term cardiac prognosis of patients with MD	MD genotype	260	108	108	[23]
TB. Haack et al. 2013	*CR*	To investigate by NGS three unrelated family	Patients with HCM, lactic acidosis, isolated complex I deficiency on muscle biopsy	5	*5*	5	[156]
E. Malfatti et al. 2013	*RS*	To determine the long-term incidence of cardiac life-threatening complications and death	MT-TL1 m.3243A>G	41	10	10	[157]
KG. Hollingsworth et al. 2012	*CCS*	To determine whether cardiac complications in patients with MD are sufficiently common to warrant prospective screening in all mutation carriers	MT-TL1 m.3243A>G	10	10	10	[158]
Z. Liu et al. 2012	*CR*	To investigate clinical, genetic and molecular characterization of a family with a likely maternally transmitted HCM	Patients with HCM	7	4	4	[106]
P. Kaufmann et al. 2011	*PS*	To describe clinical and laboratory features associated with mtDNA mutation in 35 unrelated family	MT-TL1 m.3243A>G	85	4	4	[159]
C. Nozières et al. 2011	*CR*	To investigate a patient with diabetes and hypokinetic cardiomyopathy	MD-phenotype	1	1	1	[160]
G. Limongelli et al. 2010	*RS*	To determine the frequency and natural history of cardiac disease in patients with primary respiratory chain disease	RCD phenotype	32	26	16	[53]
K. Wahbi et al. 2010	*RS*	To determine the prevalence and prognostic value of cardiac abnormalities in unrelated families with mtDNA mutation	MT-TK m.8344A>G	18	9	9	[60]
N. Yajima et al. 2009	*CR*	To investigate a patient with pericardial effusion and heart failure in whom MCM was definitively diagnosed	Patient with LVH	1	1	1	[161]
KA. Majamaa-Voltti et al. 2006	*RS*	To follow the clinical course of patients with the MD	MT-TL1 m.3243A>G	33	18	18	[162]
JH. Chae et al. 2004	*RS*	To evaluate the incidence and clinical heterogeneity of m.3243A>G mutation in Korean population	MD phenotype	85	2	0*	[163]
D. Holmgren et al. 2003	*RS*	To determine the frequency of CM in children with MD and describe their clinical course, prognosis and cardiological manifestations	Genotype or phenotype of MD	101	17	4	[164]
Y. Momiyama et al. 2003	*CCS*	To investigate the prevalence of several mtDNA mutations in diabetic patients with LVH	Patients with type 2 DM with or without LVH	168	68	38	[165]
F. Terasaki et al. 2001	*CR*	To investigate CM showing progression from the HCM to DCM	HCM phenotype	1	1	1	[68]
WS. Shin et al. 2000	*CCS*	To clarify the relationship between variation in mtDNA and the development of CM	2 brothers with maternally inherited CM, 126 HCM and 55 DCM patients	183	6	6	[166]
Y. Okajima et al. 1998	*PS*	To investigate cardiac function in children with MELAS and clarify the clinical features of CM	MELAS phenotype	11	9	9	[167]
PF. Chinnery et al. 1997	*RS*	To evaluate the relationship between the incidence of specific clinical features and the level of mutant mtDNA in blood and/or skeletal muscle	MT-TL1 m.3243A>G and or MT-TK m.8344A>G	245	3	3	[168]
R. Anan et al. 1995	*RS*	To determine the cardiac involvement in MD	MDs phenotype	17	10	10	[169]
J. Guenthard et al. 1995	*CR*	To evaluate the CM in a patient with RCD	RCD phenotype	1	1	0	[170]
E. Ciafaloni et al. 1992	*CCS*	To determine the relationship of m.3243A>G mutation to the MELAS phenotype	MELAS phenotype	23	4	4	[171]
M. Zeviani et al. 1991	*CR*	To identify a maternally inherited disorder characterized by CM	MD phenotype	21	5	5	[172]
T. Ozawa et al. 1990	*CR*	To identify mitochondrial abnormalities in CM	Cardiac tissue specimens from patients with CM	5	5	3	[173]

CCS, case-control study; CM, cardiomyopathy; CR, case report; DCM, dilated cardiomyopathy; HCM, hypertrophic cardiomyopathy; LVH, left ventricular hypertrophy; MCM, mitochondrial cardiomyopathy; MD, mitochondrial disease; MELAS, Mitochondrial encephalomyopathy, lactic acidosis, and stroke-like episodes; PS, prospective study; RCD, respiratory chain disease; RS, retrospective study. MD genotype is referred to patients with pathogenic mutations. **^§^** Positive genotype is referred to patients with mtDNA and/or nDNA mutation. * The authors do not clarify if molecular investigation was performed on blood or other tissue of CM patients.

**Table 3 ijms-22-05742-t003:** List of major mitochondrial and nuclear (mtDNA and nDNA) genes collected through literature and database research over the past ten years in association with mitochondrial heart disease.

Function	Gene OMIM ID	Gene	Transcript ID	Length bps	Protein	Inheritance
	612035	*AARS2*	ENST00000244571.5	4798	Alanyl-tRNA synthetase 2	AR
	611103	*ACAD9*	ENST00000308982.12	2445	Acyl-CoA dehydrogenase family member 9	AR
	609575	*ACADVL*	ENST00000356839.10	2184	Acyl-CoA dehydrogenase very long chain	AR
	610345	*AGK*	ENST00000649286.2	3628	Acylglycerol kinase	AR
	613183	*BOLA3*	ENST00000327428.10	555	BOLA family member 3	AR
	606158	*BSCL2*	ENST00000360796.10	1710	BSCL2 lipid droplet biogenesis associated, seipin	AR
	614775	*COA3*	ENST00000328434.8	780	Cytochrome C oxidase assembly factor 3	AR
	613920	*COA5*	ENST00000328709.8	1770	Cytochrome C oxidase assembly factor 5	AR
	614772	*COA6*	ENST00000366615.10	1741	Cytochrome C oxidase assembly factor 6	AR
	609825	*COQ2*	ENST00000647002.2	1525	Coenzyme Q2, polyprenyltransferase	AR
	612898	*COQ4*	ENST00000300452.8	1245	Coenzyme Q4	AD/AR
	614647	*COQ6*	ENST00000334571.7	2109	Coenzyme Q6, monooxygenase	AR
	601683	*COQ7*	ENST00000321998.10	2642	Coenzyme Q7, hydroxylase	AR
	606980	*COQ8A*	ENST00000366777.4	2866	Coenzyme Q8A	AR
	612837	*COQ9*	ENST00000262507.11	1630	Coenzyme Q9	AR
	602125	*COX10*	ENST00000261643.8	2898	Cytochrome C oxidase assembly factor COX10	AR
	614478	*COX14*	ENST00000550487.6	484	Cytochrome C oxidase assembly factor COX14	AR
	603646	*COX15*	ENST00000016171.6	5030	Cytochrome C oxidase assembly homolog COX15	AR
	614698	*COX20*	ENST00000411948.7	2295	Cytochrome C oxidase assembly factor COX20	AR
	124089	*COX6B1*	ENST0000649813.2	488	Cytochrome C oxidase subunit 6B1	AR
	601269	*C1QBP*	ENST00000225698.8	1169	Complement component C1q-binding protein	AR
	600650	*CPT2*	ENST00000371486.4	2699	Carnitine palmitoyltransferase 2	AD/AR
	608977	*DNAJC19*	ENST00000382564.8	1416	DNAJ heat shock protein family (Hsp40) member C19	AR
	602462	*DRP-1*	ENST00000324989.12	3174	Collapsin response mediator protein 1	-
	602292	*ECHS1*	ENST00000368547.4	1277	Enoyl-CoA hydratase, short chain 1	AR
	605367	*ELAC2*	ENST00000338034.9	3767	ELAC ribonuclease Z 2	AR
	608253	*ETFA*	ENST00000557943.6	2289	Electron transfer flavoprotein subunit alpha	AR
	130410	*ETFB*	ENST00000309244.9	872	Electron transfer flavoprotein subunit beta	AR
	231675	*ETFDH*	ENST00000511912.6	3111	Electron transfer flavoprotein dehydrogenase	AR
	609003	*FIS1*	ENST00000223136.5	870	Fission, mitochondrial 1	-
	613622	*FOXRED1*	ENST00000263578.10	1951	FAD dependent oxidoreductase domain containing protein 1	AR
	606829	*FXN*	ENST00000484259.3	6978	Frataxin	AR
	608536	*GTPBP3*	ENST00000324894.13	2566	GTP binding protein 3	AR
	600890	*HADHA*	ENST00000380649.8	2943	Hydroxyacyl-CoA dehydrogenase trifunctional multienzyme complex subunit alpha	AR
	143450	*HADHB*	ENST00000317799.10	1997	Hydroxyacyl-CoA dehydrogenase trifunctional multienzyme complex subunit beta	AR
	601421	*KARS1*	ENST00000302445.8	1991	Lysyl-tRNA synthetase 1	AR
	607544	*LRPPRC*	ENST00000260665.12	6603	Leucine rich pentatricopeptide repeat containing	AR
	608419	*MCEE*	ENST00000244217.6	824	Methylmalonyl-CoA epimerase	AR
	608506	*MFN1*	ENST00000471841.6	5212	Mitofusin 1	-
	608507	*MFN2*	ENST00000235329.10	4407	Mitofusin 2	-
	607481	*MMAA*	ENST00000649156.2	5944	Metabolism of cobalamin associated A	AR
	607568	*MMAB*	ENST00000545712.7	4090	Metabolism of cobalamin associated B	AR
	609831	*MMACHC*	ENST00000401061.9	5049	Metabolism of cobalamin associated C	AR
	611935	*MMADHC*	ENST00000303319.10	1392	Metabolism of cobalamin associated D	AR
	609058	*MMUT*	ENST00000274813.4	3811	Methylmalonyl-CoA mutase	AR
	607118	*MRPL3*	ENST00000264995.8	1708	Mitochondrial ribosomal protein L3	AR
	611849	*MRPL44*	ENST00000258383.4	1689	Mitochondrial ribosomal protein L44	AR
	605810	*MRPS22*	ENST00000680020.1	1205	Mitochondrial ribosomal protein S22	AR
	614667	*MTO1*	ENST00000498286.6	10698	Mitochondrial tRNA translation optimization 1	AR
	605299	*NCOA6*	ENST00000359003.7	7083	Nuclear receptor coactivator 6	-
	300078	*NDUFA1*	ENST00000371437.5	421	NADH:Ubiquinone oxidoreductase subunit A1	XLR
	603835	*NDUFA10*	ENST00000252711.7	4855	NADH:Ubiquinone oxidoreductase subunit A10	AR
	612638	*NDUFA11*	ENST00000308961.5	575	NADH:Ubiquinone oxidoreductase subunit A11	AR
	602137	*NDUFA2*	ENST00000252102.9	649	NADH:Ubiquinone oxidoreductase subunit A2	AR
	606934	*NDUFAF1*	ENST00000260361.9	1509	NADH:Ubiquinone oxidoreductase complex assembly factor 1	AR
	609653	*NDUFAF2*	ENST00000296597.10	632	NADH:Ubiquinone oxidoreductase complex assembly factor 2	AR
	612911	*NDUFAF3*	ENST00000326925.11	902	NADH:Ubiquinone oxidoreductase complex assembly factor 3	AR
	611776	*NDUFAF4*	ENST00000316149.8	2407	NADH:Ubiquinone oxidoreductase complex assembly factor 4	AR
	612360	*NDUFAF5*	ENST00000378106.10	5449	NADH:Ubiquinone oxidoreductase complex assembly factor 5	AR
	603839	*NDUFB3*	ENST00000237889.9	493	NADH:Ubiquinone oxidoreductase subunit B3	AR
	603842	*NDUFB7*	ENST00000215565.3	535	NADH:Ubiquinone oxidoreductase subunit B7	-
	601445	*NDUFB9*	ENST00000276689.8	691	NADH:Ubiquinone oxidoreductase subunit B9	AR
	157655	*NDUFS1*	ENST00000233190.11	11660	NADH:Ubiquinone oxidoreductase core subunit S1	AR
	602985	*NDUFS2*	ENST00000676972.1	1613	NADH:Ubiquinone oxidoreductase core subunit S2	AR
	603846	*NDUFS3*	ENST00000263774.9	894	NADH:Ubiquinone oxidoreductase core subunit S3	AR
	602694	*NDUFS4*	ENST00000296684.10	669	NADH:Ubiquinone oxidoreductase core subunit S4	AR
	603848	*NDUFS6*	ENST00000274137.10	518	NADH:Ubiquinone oxidoreductase core subunit S6	AR
	601825	*NDUFS7*	ENST00000233627.14	758	NADH:Ubiquinone oxidoreductase core subunit S7	AR
	602141	*NDUFS8*	ENST00000313468.10	737	NADH:Ubiquinone oxidoreductase core subunit S8	AR
	161015	*NDUFV1*	ENST00000322776.11	1560	NADH:Ubiquinone oxidoreductase core subunit V1	AR
	600532	*NDUFV2*	ENST00000318388.11	857	NADH:Ubiquinone oxidoreductase core subunit V2	AR
	608100	*NFU1*	ENST00000410022.7	898	NFU1 iron-sulfur cluster scaffold	AR
	139139	*NR4A1*	ENST00000394825.6	2485	Nuclear receptor subfamily 4 group A member 1	-
	613621	*NUBPL*	ENST00000281081.12	3040	Nucleotide binding protein like	AR
	605290	*OPA1*	ENST00000361510.8	6429	OPA1, mitochondrial dynamin like GTPase	AD/AR
	612036	*PARS2*	ENST00000371279.4	2356	Prolyl-tRNA synthetase 2	AR
	232000	*PCCA*	ENST00000376285.6	2484	Propionyl-CoA carboxylase subunit alpha	AR
	232050	*PCCB*	ENST00000251654.9	1799	Propionyl-CoA carboxylase subunit beta	AR
	607429	*PDSS1*	ENST00000376215.10	1584	Prenyl diphosphate synthase subunit 1	AR
	610564	*PDSS2*	ENST00000369037.9	3536	Prenyl diphosphate synthase subunit 2	AR
	174763	*POLG*	ENST00000268124.11	4462	Polymerase, DNA, gamma	AD/AR
	617209	*QRSL1*	ENST00000369046.8	4106	Glutaminyl-tRNA amidotransferase subunit QRSL1	AR
	179710	*RCC1*	ENST00000373833.10	2844	Regulator of chromosome condensation 1	-
	614917	*RMND1*	ENST00000444024.3	1948	Required for meiotic nuclear division 1 HOMOLOG	AR
	603644	*SCO1*	ENST00000255390.10	9577	Synthesis of Cytochrome C oxidase 1	AR
	604272	*SCO2*	ENST00000395693.8	984	SCO2, Cytochrome C oxidase assembly protein	AR
	600857	*SDHA*	ENST00000264932.11	2693	Succinate dehydrogenase complex flavoprotein subunit A	AR
	613019	*SDHAF2*	ENST00000301761.7	1186	Succinate dehydrogenase complex assembly factor 2	-
	602690	*SDHD*	ENST00000375549.7	1439	Succinate dehydrogenase complex subunit D	AR
	603941	*SLC19A2*	ENST00000236137.10	3612	Solute carrier family 19 member 2	AR
	603377	*SLC22A5*	ENST00000245407.8	3277	Solute carrier family 22 member 5	AR
	600370	*SLC25A3*	ENST00000228318.8	5987	Solute carrier family 25 member 3	AR
	103220	*SLC25A4*	ENST00000281456.11	4415	Solute carrier family 25 member 4	AD/AR
	613698	*SLC25A20*	ENST00000319017.5	1778	Solute carrier family 25 member 20	AR
	185620	*SURF1*	ENST00000371974.8	1092	SURF1, Cytochrome C oxidase assembly factor	AR
	300394	*TAZ*	ENST00000601016.6	1906	Tafazzin	XLR
	612418	*TMEM70*	ENST00000312184.6	2032	Transmembrane protein 70	AR
	601243	*TOP3A*	ENST00000321105.10	6596	Topoisomerase, DNA, III, alpha	AR
	611023	*TRMT5*	ENST00000261249.7	5304	tRNA methyltransferase 5	AR
	604723	*TSFM*	ENST00000652027.2	1997	Ts translation elongation factor, mitochondrial	AR
	602389	*TUFM*	ENST00000313511.8	2011	Tu translation elongation factor, mitochondrial	AR
	131222	*TYMP*	ENST00000252029.8	1586	Thymidine phosphorylase	AR
	610957	*YARS2*	ENST00000324868.13	2117	Tyrosyl-tRNA synthetase 2	AR
	516060	*MT-ATP6*	ENST00000361899.2	681	Mitochondrially encoded ATP synthase membrane subunit 6	M
	516070	*MT-ATP8*	ENST00000361851.1	207	Mitochondrially encoded ATP synthase membrane subunit 8	M
	516030	*MT-CO1*	ENST00000361624.2	1542	Mitochondrially encoded Cytochrome C oxidase I	M
	516040	*MT-CO2*	ENST00000361739.1	684	Mitochondrially encoded Cytochrome C oxidase II	M
	516050	*MT-CO3*	ENST00000362079.2	784	Mitochondrially encoded Cytochrome C oxidase III	M
	516020	*MT-CYB*	ENST00000361789.2	1141	Mitochondrially encoded cytochrome b	M
	516000	*MT-ND1*	ENST00000361390.2	956	Mitochondrially encoded NADH-ubiquinone oxidoreductase core subunit 1	M
	516001	*MT-ND2*	ENST00000361453.3	1042	Mitochondrially encoded NADH-Ubiquinone oxidoreductase core subunit 2	M
	516003	*MT-ND4*	ENST00000361381.2	1378	Mitochondrially encoded NADH-Ubiquinone oxidoreductase core subunit 4	M
	516005	*MT-ND5*	ENST00000361567.2	1812	Mitochondrially encoded NADH-Ubiquinone oxidoreductase core subunit 5	M
	516006	*MT-ND6*	ENST00000361681.2	525	Mitochondrially encoded NADH-Ubiquinone oxidoreductase core subunit 6	M
	561000	*MT-RNR1*	ENST00000389680.2	954	Mitochondrially encoded 12S rRNA	M
	561010	*MT-RNR2*	ENST00000387347.2	1559	Mitochondrially encoded 16S rRNA	M
	590025	*MT-TE*	ENST00000387459.1	69	Mitochondrially encoded tRNA glutamic acid	M
	590070	*MT-TF*	ENST00000387314.1	71	Mitochondrially encoded tRNA phenylalanine	M
	590035	*MT-TG*	ENST00000387429.1	68	Mitochondrially encoded tRNA glycine	M
	590040	*MT-TH*	ENST00000387441.1	69	Mitochondrially encoded tRNA histidine	M
	590045	*MT-TI*	ENST00000387365.1	69	Mitochondrially encoded tRNA isoleucine	M
	590060	*MT-TK*	ENST00000387421.1	70	Mitochondrially encoded tRNA lysine	M
	590050	*MT-TL1*	ENST00000386347.1	75	Mitochondrially encoded tRNA leucine 1	M
	590055	*MT-TL2*	ENST00000387456.1	71	Mitochondrially encoded tRNA leucine 2	M
	590010	*MT-TN*	ENST00000387400.1	73	Mitochondrially encoded tRNA asparagine	M
	590075	*MT-TP*	ENST00000387461.2	68	Mitochondrially encoded tRNA proline	M
	590005	*MT-TR*	ENST00000387439.1	65	Mitochondrially encoded tRNA arginine	M
	590090	*MT-TT*	ENST00000387460.2	66	Mitochondrially encoded tRNA threonine	M
	590105	*MT-TV*	ENST00000387342.1	69	Mitochondrially encoded tRNA valine	M
	590095	*MT-TW*	ENST00000387382.1	68	Mitochondrially encoded tRNA tryptophan	M

AD: autosomal dominant; AR: autosomal recessive; XLR: X-linked recessive; M: Matrilineal. The colored square indicates the gene function as showed in Figure 1. Data from References [7,16,19,48,151,152,154,155,174,175,176]. The Transcript ID refers to RefSeq with NM number (from Ensembl: Genome browser for vertebrate genomes. Available on line: https://www.ensembl.org/index.html database; accessed 2 October 2020).

## Data Availability

Data is contained within the article or Appendix A.

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
