# Peer review of "Molecular Epidemiology of Mitochondrial Cardiomyopathy: A Search Among Mitochondrial and Nuclear Genes"

_ijms, 2021, doi:10.3390/ijms22115742_

Round 1

Reviewer 1 Report

The authors present here a review summarizing the latest known evidences on cardiomyopathies linked to mitochondrial diseases (MD), and highlight the importance of precision diagnosis based on Next Generation Sequencing. This topic is quite recent and of major importance. 

However, the manuscript organization and paragraph content is a bit heterogeneous and the overall plan has to be designed again. Editing of English language and style is required.

Overall, the manuscript would like to match the title : Mitochondrial cardiomyopathy: so many genes and one disease? Next generation sequencing to make molecular diagnosis, but misses the target.

Although one should recognize the important work that was realized to gather all the written information, the manuscript mixes a review article (first part) + builds hypothesis (mutation occurring on some mitochondrial genes, could happen on the 130 identified genes) + bring solutions (NGS) to an hypothetical issue.

Some paragraphs brings too much details that does not add to the main idea: e.g From line 38 to 225, the reader may understand the concept but the text is wordy and bring too many details (e.g line 119 to 130). Some are not relevant (line 133: 7 common bases... out of more than 1000 bases / Line 629-635: 02 October 2020 repeated 7 times)).

Line 53 100.000 instead of 100,000 (again lane 247)

Line 81: not sure of the meaning

Line 340: KATP

Word splitting at the end of the lines are sometimes inappropriate (line 361, 367,  381, 436, etc).

Line 443: equipe ?

Line 504 ??

Line 596-597 : please correct the (¨) appropriate places

Line 317 Meaning of ''Which CoQ10'' ?

Line 707 Table 2: AARS2 gene is NOT 3854 bps long. Why did the authors choose this length? Why did the authors focused on one particular  transcript, when AARS2 has 4 splicing isoforms ? (Same thing applies for all other identified genes, including the transcript length)

Reviewer 2 Report

The manuscript by Mazzaccara et al., entitled “Mitochondrial cardiomyopathy: so many genes and one disease? Next-generation sequencing to make molecular diagnosis”, is a review of the phenotypes and genes associated with mitochondrial diseases.  This review includes extensive background information on mitochondrial disease and associated cardiomyopathies.  The authors also perform an extensive literature search to identify genes associated with mitochondrial disease that could be used as target genes for panel-based diagnostic sequencing.  The review is very long and the main data (lit review) is not introduced until the very end.  This reviewer recommends condensing some of the background and expanding the literature review discussion and the explanation of the potential gene panel.

Major/minor concerns”

  1. The review is very long and should be carefully read for areas that can be trimmed and consolidated.
  2. There are a number of instances where the authors provide blocks of information that could be put into supplemental tables – for example line 267, line 574, line 628.
  3. Provide a table with type of cardiomyopathy, phenotypes, class of genes, syndromes, age of onset, etc. This table will allow for a reduction in the amount of text.
  4. Please describe the molecular/cellular phenotype of the mitochondria for each cardiomyopathy.
  5. Move the methods to a supplement or the end.
  6. I disagree that WES is low coverage, a number of studies have shown that WES and WGS provide ample coverage for rare variant identification.
  7. The authors provide 130 genes that should be included in a clinical panel. What is the support for each gene for pathogenesis? How likely is it that a variant in any of these genes would meet the ACMG criteria for pathogenicity? If all variants would be VUS, are these clinically useful?
  8. Paragraph ln 574 – please add gnomAD.
  9. Line 504 – please fix format.
  10. Line 597 – variants of uncertain significance are commonly referred to as VUS.
  11. Line 101 – please clarify what the authors mean by 1500 nuclear genes and 37 mitochondrial genes – are these total for function.
  12. Please clarify line 80 - what does the 40-60% represent?

Round 2

Reviewer 1 Report

The authors adressed all the issues and comments that were pointed out.

The review is now suitable for publication.